# Myocardial Work: Methodology and Clinical Applications

**DOI:** 10.3390/diagnostics11030573

**Published:** 2021-03-22

**Authors:** Konstantinos Papadopoulos, Özge Özden Tok, Konstantina Mitrousi, Ignatios Ikonomidis

**Affiliations:** 1European Interbalkan Medical Center, Echocardiography Laboratory, 57001 Thessaloniki, Greece; ntinamitrousi@yahoo.gr; 2Cardiology Department, Memorial Bahcelievler Hospital, 34180 Istanbul, Turkey; ozgeozdenctf@hotmail.com; 3Echocardiography Laboratory, 2nd Cardiology Department, Medical School, National and Kapodistrian University of Athens, Attikon University Hospital, 12462 Athens, Greece; ignoik@gmail.com

**Keywords:** LV performance, myocardial work, speckle tracking, strain imaging

## Abstract

A precise and accurate assessment of left ventricular (LV) contractility is of utmost importance in terms of prognosis in most cardiac pathologies. Given the limitations of ejection fraction (EF) and global longitudinal strain (GLS) due to their load dependency, a novel imaging tool called myocardial work (MW) has emerged as a promising method for LV performance evaluation. MW is a novel, less load-dependent method based on computation of myocardial strain–arterial blood pressure curves. This method provides a more detailed assessment of segmental and global LV function incorporating the patient’s LV pressure and is derived by brachial artery pressure utilizing an empiric reference curve adjusted to the duration of the isovolumic and ejection phases as determined by echocardiography. The clinical implications of this unique method have been expanding in the last few years, which attest to the robust additive role of MW in routine practice.

## 1. Introduction

Left ventricular performance has traditionally been assessed by ejection fraction (EF), which has been demonstrated to be a prognostic parameter in numerous studies [1,2,3]. However, EF is notably subjective, with various limitations [4] and wide interobserver variability. Echocardiography has constantly evolved with the help of new hardware and software systems. Speckle tracking imaging with global longitudinal strain (GLS) is increasingly utilized to assess even subtle myocardial dysfunction as this is a less angle- and operator-dependent method. Although it is a novel and well-validated method for clinical utility in the assessment of cardiac diseases, it remains limited by its load dependency [5,6]. Increasing the afterload may decrease GLS and lead to false conclusions about LV contractility. A meta-analysis of 24 studies showed that variations in afterload and blood pressure can affect the normal range of strain values [7].

Following this concept, a new echocardiographic tool called myocardial work (MW), which measures the LV pressure–strain relationship through a noninvasive method, has evolved. Work is calculated as force applied over length, while myocardial work is calculated as LV pressure applied over strain, which explains why this method does not calculate the actual LV work but is a valid approximation of it. Russel et al. were the first to develop this tool to assess segmental and global myocardial work by introducing blood pressure with a noninvasive method. In their study, the LV pressure–volume loop correlated well with the pressure–strain loop (PSL) in patients with left bundle branch block (LBBB) and dyssynchrony as well as in patients with ischemic heart disease [8,9]. The results also demonstrated strong correlation and agreement with the segmental glucose metabolism measured by fluorodeoxyglucose positron emission tomography (FDG-PET).

Traditionally, LV work is evaluated by the pressure–volume relationship, which reflects the myocardial oxygen consumption and eventually the LV performance [10]. The pressure–volume loop, however, is measured with invasive methods, which explains the lack of use of this method in routine clinical practice. MW is an alternative tool to assess cardiac mechanics and a less load-dependent, noninvasive method for LV performance evaluation as it incorporates both LV afterload and deformation. MW has already been validated in coronary artery disease and heart failure patients undergoing cardiac resynchronization therapy [11,12,13,14].

The aim of this review is to meticulously describe how to assess MW and all its components in a step-by-step fashion, refer to the normal reference values that have been proposed in previous published studies, and discuss the clinical implications of this method.

## 2. Myocardial Work Methodology

LV MW is a novel, speckle tracking-based method that evaluates LV work and is estimated by employing brachial artery blood pressure and LV GLS. The profile of the estimated LV pressure curve is calculated using an empiric reference pressure curve that is adjusted according to the duration of the isovolumic and ejection phases as determined by echocardiography. MW is measured from the PSL areas that are constructed from the LV pressure curves combined with strain GLS (Figure 1). The steps of MW and its components, namely global work index (GWI), global constructed work (GCW), global wasted work (GWW), and global work efficiency (GWE), are as follows. First, all apical transthoracic views (apical long axis, four chambers (4C), and two chambers (2C)) are acquired, and the GLS and bull’s eye of the LV are estimated by speckle tracking echocardiography. Next, the timing of the valvular events is determined by visualizing the opening and closure of aortic and mitral valves from the three-chamber apical view of LV and by placing a cursor corresponding to MV opening (MVO) and closure (MVC) and aortic valve opening (AVO) and closure (AVC) along the RR interval of the accompanying electrocardiogram (ECG) trace. Finally, the patient’s blood pressure is measured with a simple brachial cuff (Figure 2), and both the systolic and diastolic components are inserted into the application. Peak systolic LV pressure is considered to be equal to peak arterial pressure in the absence of a gradient through the left ventricular outflow tract (LVOT) and the aortic valve.

Attention should be given when calculating LV GLS as there are a few details that may affect the measurement of MW. All apical images should be recorded in 40–80 frames per second (fps) with clear visualization of the endocardial border, and the region of interest (ROI) width should be adjusted in order to cover the whole myocardium and not the pericardium. If the pericardium is included, MW may be underestimated. Conversely, if the ROI width is too narrow, then it is the endocardium that is mainly included, which leads to overestimation of MW [15].

After performing all these steps, the system automatically provides bull’s eye of the global myocardial work index and global work efficiency of the LV with all 17 segments and all values of MW components. Additionally, the application demonstrates the pressure–strain loop that is equal to the GWI of the LV. By choosing a specific segment, we have the ability to compare the global and segmental WI along with their PSL at the same time. The application utilizes the previously defined event timing of the valve opening and closure on the pressure–strain loop and thus separates the different phases of the cardiac cycle isovolumic contraction, ejection time, isovolumic relaxation, and diastolic filling (Figure 1). In the same screen, all MW values and bars indicating the relationship between constructive and wasted work (global and segmental) are included.

GWI is calculated from mitral valve closure to mitral valve opening, which is the mechanical systole including isovolumic contraction. Constructive work is the work performed during systolic shortening, plus the negative work performed during lengthening in isovolumetric relaxation. The wasted work is the negative work performed during lengthening in systole, plus the work performed during shortening in isovolumetric relaxation. The global work efficiency is the constructive work divided by the sum of constructive and wasted work (Table 1). All these data are calculated automatically after completion of the given steps that are required for this method. This application is advantageous as it incorporates blood pressure with a noninvasive method that is measured by a simple brachial cuff, which makes the method practical and easy to use in daily practice in echocardiography laboratories. All MW components apart from GWE have the same units and are expressed in mmHg% as they reflect the power over the cardiac cycle longitudinal strain. GWE is expressed in % as it is the fraction of GCW divided by the sum of GCW + GWW.

## 3. Normal Reference Values

In order to define a parameter as abnormal, a good description of “normal” is essential. Because MW has been developed recently, there have only been a few studies that have included healthy subjects and described the normal values of MW and all of its components. The NORRE study [16] was the first large, multicenter, prospective study that included 226 middle-age healthy subjects. MW was analyzed and normal reference values were determined according to age and sex. Furthermore, in a subanalysis of this study [17], a direct comparison between MW components and conventional 2D EF and speckle tracking echocardiography with GLS was given. As expected, there was a good correlation between the EF, GLS, and MW indices. In particular, on multivariable analysis, GWI was significantly correlated with GLS (*p* < 0.001), EF (*p* = 0.02), systolic blood pressure (SBP) (*p* < 0.001), and global radial strain (GRS) (*p* = 0.004). GCW correlated well with GLS (*p* < 0.001), SBP (*p* < 0.001), GRS (*p* = 0.02), and global circumferential strain (GCS) (*p* = 0.01). The reference values from the NORRE study are used in routine practice in most echocardiography departments as normal reference values. Another study with healthy subjects from the STAAB cohort [18] analyzed MW components according to age and sex and compared them with conventional echocardiography parameters, such as EF and GLS. In this study, all values apart from GWI were independent of sex and stable till the age of 45 years. GWI was higher in women when compared to men, while GCW, GWE, and GWW did not differ in women and men. With increasing age, GWW also increased, resulting in decreased GWE, while GWI and GCW showed a modest increment at the age of 45 but not in older patients. All MW indices were associated with EF and GLS, but the correlation was weak (Kendall’s τ was 0.13–0.18 for EF and 0.09–0.44 for GLS), suggesting that this new method is more accurate in estimating the LV performance and perhaps detects subclinical dysfunction that previous methods could not demonstrate. Galli et al. [19] also performed advanced analysis of healthy subjects and described the values of MW components according to age, gender, and the LV territory. They found no differences between the age-specific subgroups, while statistically significant difference was demonstrated between men and women; in particular, GWI and GCW were higher in females (2031 ± 247 vs. 1874 ± 232 mmHg%, *p* = 0.001 and 2289 ± 261 vs. 2194 ± 207 mmHg%, *p* = 0.04, respectively). This is in agreement with the established knowledge that GLS and E’ are much higher in women than in men. In this study, Galli et al. also demonstrated a significant difference between basal, mid, and apical segmental LV work, increasing from base to apex. Likewise, previous studies have demonstrated that segmental strain increases from base to apex [20], which is attributed to the curvature and architecture of the apical LV fibers. Table 2 refers to all normal reference values described by the three aforementioned studies.

## 4. Myocardial Work in Cardiac Disease

Estimation of LV systolic function is essential in all cardiac diseases, and EF continues to be the most commonly used parameter. Although speckle tracking echocardiography with GLS provides more detailed information about global and segmental systolic function, giving the opportunity to detect subclinical dysfunction, it remains load-dependent [6]. MW, on the other hand, is a less load-dependent tool for LV function evaluation as it incorporates the LV afterload. MW gives a rough estimation of the work that every segment of the LV produces during the cardiac cycle. This work is influenced by the power of the contraction of myocardial fibers, LV loading conditions, and the wall stress applied on the LV segments. Loading conditions include both preload and afterload and play an important role in LV contraction. MW includes afterload in its equation on the basis that aortic pressure is equal to systolic LV pressure in the absence of LVOT and aortic gradients. However, even with those limitations, MW has been found to be a more sensitive index of segmental and global LV performance compared to EF and GLS. This was confirmed in a previously published study that included hypertensive patients. It was demonstrated that, in acute pressure overload, EF remains unaffected and GLS decreases while MW indices show an increase [21]. In these patients, work increases as every segment must provide more energy to pump against the raised afterload. However, in patients with chronic pressure overload, when myocardial remodeling and hypertrophy appears, tissue fibrosis is more pronounced and correlates with the inability of the hypertrophied segments to increase their work during an acute pressure overload. MW indices will be lower in this subgroup of patients [21]. These results show that MW is more sensitive in detecting segmental tissue fibrosis even in the early stages and can add valuable information when measured alongside EF and GLS. 

In a normal heart, all 17 segments contract at the same time and produce very little wasted work. In diseased hearts with dyssynchronous contraction of LV, bundle branch block, ischemic, or dilated cardiomyopathy, there are segments that lengthen during systole or contract against a closed aortic valve, producing a lot of wasted work. (Figure 3) This explains why MW is much more sensitive in detecting the actual LV efficiency even in normal contracting ventricles. Wasted work does not contribute to LV ejection, and healthy hearts produce an insignificant amount of wasted work [4]. MW has already proven to be more sensitive than EF in detecting LV dysfunction in cases of cardiomyopathy and normal EF [22] in patients with chronic kidney disease where subclinical dysfunction can be detected [23]. Similarly, in athletes with normal cardiac function, MW can detect myocardial changes after long-term intensive exercise [24]. Several patterns for bull’s eye and PSL have been published according to the underlying pathology, showing the difference between normal subjects and patients with hypertension, ischemic, and nonischemic cardiomyopathy [25] (Figure 4 and Figure 5).

In the COVID-19 pandemic era, MW has shown better sensitivity in detecting cardiac complications from coronavirus [26]. These complications vary from acute myocardial infarction, myocarditis, pericarditis, and heart failure. There are case reports where patients with normal EF and GLS without any wall motion abnormalities showed significantly reduced GWI that recovered to near-normal in follow-up echo exams [27]. MW in this subgroup of patients again detected myocardial damage far before it could be with eyeball assessment, EF, or GLS. However, a large multicenter study including thousands of patients with COVID-19-related cardiac complications must be conducted in order to prove the additive value of MW in detecting subclinical myocardial dysfunction.

## 5. Cardiac Dyssynchrony

MW was first described and validated in patients undergoing cardiac resynchronization therapy (CRT) [8]. The dyssynchronous ventricular septum produces large amounts of wasted work, and consequently, after response to this treatment, WW reduces and global work index and LV performance increases [8]. A recent study showed that there is acute redistribution of segmental work between septal and lateral work, which is a strong determinant of LV reverse remodeling [28].

CRT is an important treatment option in heart failure patients, although with limited indications as devices are roughly implanted in patients with bundle branch block (preferably LBBB) and EF < 35% [29]. However, even if a patient meets these criteria for CRT implantation, there is a 30–40% chance of this subgroup having no clinical improvement. These patients are called “non-responders”. There are certain theories as to why such a significant percentage of patients do not respond to treatment; however, so far, all echocardiographic parameters have failed to predict who will benefit from CRT devices. It has been shown that use of MW is increasing in patients who have responded well to CRT devices [30]. Apart from that, MW has also been used as a predictor of CRT responders. The measurement of septal wasted work along with wall motion score index (WMSI_ was proposed by Vecera et al. [31] to be a reliable parameter for predicting responders. In multiple linear regression analysis, they showed that the combination of septal WW and global WMSI predicted ESV reduction (septal wasted work, β = 0.14, *p* = 0.004; WMSI, β = 1.32, *p* = 0.0012). In area under the curve (AUC) analysis, the combination of septal WW and WMSI showed an AUC of 0.86 (0.71–1.0) for CRT response prediction. However, larger studies need to be conducted to establish this hypothesis. Aalen et al. demonstrated that the work difference between septal and lateral wall, combined with the viability detected by cardiac magnetic resonance (CMR), are the parameters to predict CRT responders, with AUC of 0.88 (95% CI: 0.81–0.95), sensitivity of 86%, and specificity of 84% [32]. In particular, the threshold for CRT response calculated average work difference of more than 900 mmHg%. Galli et al. further investigated the effect of preserved GCW in echocardiographic response and mortality after CRT implantation. In particular, this study showed that a GCW > 888 mmHg% was correlated with 15% ESV reduction (AUC: 0.71, 95% CI: 0.60–0.82, *p* = 0.007) and statistically significant reduction in mortality (hazard ratio (HR): 4.76, 95% CI: 1.33–17.12, *p* = 0.01) compared to patients with lower values [33]. In two other studies, Galli et al. [11,12] also showed that GCW < 1057 mmHg% identified 85% of non-responders with a positive predictive value of 88–100% but with low sensitivity and negative predictive value. In multivariate analysis septal flash, GCW > 1057 mmHg% and GWW > 384 mmHg% were the only significant predictors of CRT-positive response.

MW has been validated in patients with LBBB undergoing CRT implantation from the very beginning, and since then, all the studies that have been conducted have been convincing about the role of the method in this subgroup of patients. The correlation of MW with mortality rates and prognosis will play an important role in the establishment of the method. Mostly segmental, but also global, values of MW indices, especially wasted and constructive work, should be routinely used when CRT patients are under investigation. However, treatment still cannot be guided by only this echocardiographic parameter. Clinicians should rely on symptoms and ECG findings according to guidelines until large studies have established MW as a strong indicator for CRT implantation.

## 6. Heart Failure

As MW is an index of LV performance, we expected it to be investigated in patients with heart failure. However, there have only been a few studies with limited number of patients that have shown MW to be significantly correlated with other echocardiographic and clinical parameters. Hedwig et al. demonstrated a strong correlation between GWI and N-terminal pro-B-type natriuretic peptide (NT-proBNP), cardiopulmonary exercise, and established prognostic echocardiographic parameters, such as ESV and EF. In particular, GWI < 500 mmHg% was a predictor of severely impaired LV function, inadequate cardiopulmonary exercise, and increased NT-proBNP levels, all of which are markers of poor prognosis. Patients with GWI < 500 mmHg% had a mean LV end-diastolic volume (LVEDV) of 286.1 ± 100.8 mL, LV ejection fraction (LVEF) of 21.3 ± 5.7%, and stroke volume (SV) of 45.9 ± 11.6 mL, while patients with GWI > 1000 mmHg% had a LVEDV of 147.9 ± 39.6 mL, LVEF of 42.6 ± 4.8%, and SV of 70.9 ± 14.3 mL. GWI also showed significant correlation with peak oxygen consumption (peak VO_2_) (*r* = 0.521; *p* < 0.001) and with NT-pro-BNP levels (*r* = 0.635, *p* < 0.001). Patients with a GWI of < 500 mmHg% had a significantly higher NT-pro-BNP (median 2415 pg/mL [IQR 1071, 5933]) and a lower peak VO_2_ (9.5 mL/min/kg ± 2.6) compared to patients with a GWI of > 1000 mmHg% (NT-pro-BNP median, 253 pg/mL [IQR 150, 549]; peak VO_2_ 15.6 ± 4.2 mL/min/kg) [34]. Shrub et al. analyzed patients with dilated cardiomyopathy (DCM) and found that septal WE was the only predictor of exercise capacity in patients with LV dyssynchrony (LV-DYS). In more detail, LV-DYS was associated with lower EF, lower global and septal WE, and higher global and septal WW. In patients with LV-DYS, septal WE was the only predictor of exercise capacity in multivariable analysis (β = 0.68, *p* = 0.03), whereas LVEF (β = 0.47, *p* = 0.05) and age (β = −0.42, *p* = 0.04) were predictors of exercise capacity in patients without LV-DYS [35]. In contrast to the aforementioned studies, Kosmala et al. analyzed heart failure with preserved EF (HFpEF) patients and showed that the exertional increase of GCW after treatment with spironolactone was able to predict increase in exercise capacity in this subgroup of patients [36]. The response to sacubitril/valsartan in heart failure-reduced ejection fraction (HFrEF) patients was also investigated by Bouali et al. Sacubitril/valsartan significantly increased myocardial constructive work (1023 ± 449 vs. 1424 ± 484 mmHg%, *p* < 0.0001) and myocardial work efficiency (87 (78–90) vs. 90 (86–95), *p* < 0.0001). After correction for LV size, EF, and WE, GCW was the only predictor of major adverse cardiovascular events (MACE) (HR: 0.99 (0.99–1.00), *p* = 0.04). GCW < 910 mmHg identified patients at particularly increased risk of MACE (HR: 11.09 (1.45–98.94), *p* = 0.002, log-rank test *p* < 0.0001) [37]. MW and in particular GWI, GCW, and GWW were also investigated in diabetic patients with reduced LV contractility. It was demonstrated that patients under combined therapy with GLP-1RA + SGLT-2i (glucagon like, peptide-1 receptor agonists + sodium glucose cotransporter 2 inhibitors) who had baseline LVEF < 55% showed greater improvement in GWI (*p* = 0.0037), GCW (*p* = 0.025), and GWW (*p* = 0.047) than insulin treatment or each medication alone 12 months after treatment [38].

Even though the results from the aforementioned studies are promising, all of them are of low value and cannot provide safe conclusions about the use of MW in HF patients. In a recently published study, for the first time, Wang et al. included 508 HF patients with EF < 40%. Although this was a single-center study, the results should be taken into serious consideration as MW was associated with the prognosis of patients. MW correlated with HF hospitalizations and all-cause mortality. In particular, patients with GMW < 750 mmHg% were associated with a significantly higher risk of all-cause death and HF hospitalization (HR: 3.33, 95% CI: 2.31–4.80) than patients with GMW > 750 mmHg%. These results expand the use of MW in patients with EF <40% and provide safe use of MW for not only segmental but also global values instead of EF and GLS for estimation of patient survival and rates of future hospitalization [39].

## 7. Cardiomyopathies

MW can also be beneficial in evaluation of cardiomyopathies. In a previous study, it was demonstrated that mean GCW (1722 ± 602 vs. 2274 ± 574 mmHg%, *p* < 0.001), GWE (93% (89–95%) vs. 96% (96–97%), *p* < 0.001), and GWI (1534 ± 551 vs. 1929 ± 473 mmHg%) were significantly reduced, while GWW (104 mmHg% (66–137 mmHg%) vs. 71 mmHg% (49–92 mmHg%), *p* < 0.001) was increased in hypertrophic cardiomyopathy (HCM) patients. Segmental differences of CW were observed among different phenotypes of HCM, and GCW was associated with adverse events [40]. A median value of GCW > 1730 mmHg% was related with better, event-free survival in this subgroup of patients than in those with GCW < 1730 mmHg% (*p* < 0.001). Galli et al. also found that GCW was significantly reduced in HCM (1599 ± 423 vs. 2248 ± 249 mmHg%, *p* < 0.0001). No difference was observed in GWW (141 ± 125 vs. 101 ± 88 mmHg%, *p* = 0.18) and EF (63 ± 13 vs. 6 ± 4% *p* = 0.17) between the two groups. GCW was the only predictor of myocardial fibrosis as confirmed by CMR, and in particular, a value of 1623 mmHg% showed good sensitivity and decent specificity (82 and 67%, respectively) [22]. Similar results were observed in a study with cardiac amyloidosis (CA) patients [41]. All MW values were significantly reduced compared to healthy subjects (*p* < 0.001 for all comparisons), and the average increase of GWI in exercise was more pronounced in controls (1974 mmHg% (95% CI: 1699–2250 mmHg%; *p* < 0.0001)) than in CA patients (496 mmHg% (95% CI: 156–835 mmHg%; *p* < 0.01)) and in the basal segments of LV than in the apical ones (*p* < 0.001). Furthermore, GWI identified patients with poor exercise capacity and increased levels of surrogate prognostic markers [41].

When evaluating MW components in patients with remodeled thickened ventricles, we can assume that the calculated work may not be accurate as the algorithm does not account for geometric varieties. In patients with asymmetric hypertrophy, MW will also show inhomogeneous pattern of distribution [42]. In this concept, segmental values of MW, rather than global ones, seem more accurate in fibrosis detection and assessment of hypertrophied ventricles. However, Hiemstra et al. demonstrated the additive value of global measurements in HCM with different phenotypes and correlated it with survival [40,43]. This means that MW is feasible and reliable even when there is severe remodeling of the left ventricle. It may not reflect the actual work of the LV as it does not include the wall stress and the work produced in the circumferential direction. Although it may not replace EF and GLS in standard LV evaluation, it can be used for further determination of patients who will develop major adverse events and have poor prognosis.

## 8. Coronary Artery Disease

Assessment of patients with coronary artery disease is challenging when there are no wall motion abnormalities (WMA). Echocardiography relies on subjective methods with eyeball assessment of segmental and global myocardial dysfunction and ischemia detection. Even in stress echocardiography, there is no objective measurement that can prove ischemia rather than evaluation of WMA. MW is a promising tool that has demonstrated better sensitivity and accuracy compared to EF and GLS in detecting patients with single- or multivessel coronary artery disease (CAD). GWI was found to be significantly reduced in severe CAD patients compared to those without CAD (*p* < 0.001), and it was the most powerful predictor of significant CAD (AUC: 0.786). A cut-off value of 1810 mmHg% predicted significant CAD with a sensitivity and specificity of 92 and 51%, respectively [44]. Meimoun et al. demonstrated that constructive work was the best parameter to predict segmental and global LV recovery after anterior ST elevation myocardial infarction (STEMI) and that patients with in-hospital complications had more severely impaired GCW (*p* < 0.01) [45]. In STEMI patients treated with percutaneous coronary intervention (PCI), lower values of global and segmental work index in the territory of the culprit lesion were associated with early remodeling of the LV [46]. Finally, MW has been applied in stress echocardiography examinations revealing statistically significant segmental decrease of work index in cases of inducible ischemia and inability of GWI increase, which was in contrast to healthy subjects where GWI increased by 54% [47].

## 9. Valvular Heart Disease

MW has been assessed in valvular heart disease with promising results. Jain et al. [48] added the mean aortic gradient in the blood pressure in aortic stenosis (AS) patients as a better approach for estimation of the LV afterload and demonstrated good correlation with invasively measured LV systolic pressure. In the study, they also found that after transcatheter aortic valve implantation (TAVI), MWI was reduced immediately after the intervention. This represented the reduced afterload and LV work and oxygen demand needed because AS had been treated.

In another study, in AS patients, it was demonstrated that HF symptoms correlated well with the MW values. New York Heart Association (NYHA) III–IV patients had more impaired GWI and GCW [49], which reflected the extent of tissue fibrosis due to chronic pressure overload. In the same study, the authors combined mean aortic pressure and blood pressure to define the afterload and compared this result to invasively measured LVSP. Interestingly, they found better agreement between invasive and echocardiographic measurement of afterload with mean aortic pressure (interclass correlation coefficient (ICC): 0.846; 95% CI: 0.781–0.891; *p* < 0.001) rather than peak aortic pressure (ICC: 0.772; 95% CI: 0.397–0.892; *p* < 0.001), which overestimated the LVSP. The authors concluded that MW is feasible and reliable in AS patients.

Even though these studies show promising data for evaluation of AS patients using MW, the results should be taken into careful consideration as Russel et al. [8] incorporated SBP in the MW equation only in patients without outflow gradients. The proposed protocol of corrected MWI by adding the mean aortic gradient in SBP seems feasible and reliable, but it must be applied in a large series of TAVI patients for validation and inclusion in routine practice.

MW has also been evaluated in HFrEF patients with concomitant severe functional mitral regurgitation (MR) treated with MitraClip [50]. The baseline GCW was the only statistically significant predictor of LV reverse remodeling one year after MitraClip implantation. Considering that all HF treatments, whether pharmacological or device-based (e.g., CRT), aim to reduce LV volumes and thus increase life expectancy, it is of great importance to be able to differentiate which patients will benefit from this expensive method. Particularly, in this study, reduction of diastolic volumes (ΔLVEDV > 20%) was observed in patients with better LV contractility as evaluated by EF (*p* = 0.03) and GLS (*p* = 0.01), severe MR with effective regurgitant orifice area (EROA) > 30 mm^2^ (*p* = 0.02), and preserved LV performance with higher GWI (*p* = 0.006) and GCW (*p* = 0.003). However, when ΔLVESV was evaluated, as the true indicator of reverse remodeling, GCW was the only parameter that predicted reverse remodeling. A cut-off value of GCW > 846 mmHg% was associated with 10% LVESV reduction with a sensitivity and specificity of 79 and 74%, respectively [50]. Again, as the sample of the study was small, MW cannot be applied in daily practice in such patients except for research purposes. There is growing debate about the effectiveness of MitraClip implantation in HFrEF patients and secondary MR after the announcement of MITRA-FR and COAPT trials [51,52,53]. The term proportionate and disproportionate MR has emerged [54] to detect “responders” in this treatment in terms of clinical and echocardiographic improvement, but further research needs to be done. We consider that, in the future, indications for implanting MitraClip in this subgroup of patients will demand differentiation of HF symptoms due to myocardial fibrosis from those who have severe MR and will benefit from MitraClip. Speckle tracking echocardiography and MW may play important role in this direction.

## 10. Right Ventricle Assessment

MW has already been investigated recently in right ventricle (RV) and seems to correlate better than other conventional echocardiography parameters with invasive stroke volume and stroke volume index [55]. However, this method needs further research and development to estimate the real global RV performance because we are at the moment restricted to include only the segments that we can appreciate from 4C view; 4D echocardiography may give an answer to this problem.

## 11. Limitations and Future Directions

Traditionally, LV stroke work is the amount of energy that the heart converts to work during systole when pumping blood into the aorta. LV work is measured invasively with the pressure–volume loop, which demonstrates changes in intraventricular volume and pressure during a normal cardiac cycle. This work is affected by the size of the ventricle and both the preload and afterload. In large ventricles, the volume increases and the ventricle consequently contracts with greater pressure. Preload is important in assessing the contractile properties of the ventricle as it reflects the tension of the muscle when it starts to contract. Afterload is also important as it represents the work that the ventricle must generate to pump the blood to the aorta during systole.

Myocardial work was developed as an index of the work performed by the myocardium as it ejects blood during systole and is considered as a less load-dependent method compared to EF and GLS for assessment of LV performance. The protocol of this method involves only the afterload and in particular the systolic aortic pressure, which is equal to the systolic LV pressure in the absence of LVOT and aortic valve gradients. It needs to be mentioned that the protocol does not include the afterload that comes from arterial stiffness, vascular resistance, and reflection waves. In aortic stiffness, reflection waves come earlier into the systole and augment the afterload. This is closely related to LV diastolic dysfunction, ischemia, increased LV mass, and decreased deformation (GLS) [56]. The pressure–volume loop represents the ventricular–arterial coupling (VAC), which is constantly changing to match ventricular end-systolic and arterial elastances. Because MW is an index of LV stroke work represented by the pressure–strain loop, it should routinely involve the afterload generated by the arterial tree. There are several ways to measure aortic stiffness (central systolic and diastolic blood pressure instead of brachial blood pressure, pulse wave velocity, arterial impedance (Zca), and valvulo-arterial impedance (Zva)) that should be considered for inclusion in the MW methodology as a more complete assessment of afterload.

Even if MW measures the LV work during systole, end-diastolic LV pressure, which reflects the preload, is also important in LV work assessment. MW takes as granted that, in normal hearts, LVEDP is very low (2–3 mmHg) and does not affect the overall pressure–strain loop area. This means that the method should be restricted to patients with normal functional state of circulation and normal filling pressures of the ventricle. In the future, preload should be taken into account in the MW methodology in order to include all patients suffering from right ventricular pressure and volume overload conditions. Atrial fibrillation should also be taken into consideration as preload changes and MW may lead to inconclusive results from cycle to cycle. For the moment, as MW agrees with invasive methods for measuring LV stroke work and has been validated under several conditions, we accept the method as a more precise index of LV systolic performance. We should not forget that a meta-analysis of 24 studies showed alterations of GLS values in different afterload conditions [7]. As GLS is a reliable index for LV contraction, by incorporating only the afterload in the equation, a more detailed tool can be created. Of course, it cannot replace EF and GLS but can increase the accuracy in overall LV assessment.

AS and LVOT obstruction (LVOTO) is another contraindication for this method. The addition of the mean transaortic gradient to the brachial systolic pressure in AS cases has not been extensively validated. Even though several small studies showed good correlation with invasive measurements, we can only assume that this practice is correct. In HCM with LVOTO, there are no available data, so MW is not recommended for LV performance evaluation.

MW does not take into account anatomic variations, which may lead to limitations. Wall thickness, LV curvature, and radius are not accounted for in the protocol as they may alter the wall stress applied on the segments and consequently the segmental and global work. 4D application that includes all myocardial layers from endocardium to epicardium, LV mass, work produced from circumferential direction, and area strain may eliminate false results that come from geometric assumptions. Patients with normal ventricles without concentric or eccentric hypertrophy and severe remodeling are at the moment considered to be ideal candidates for application of this method.

Further development of the method to enable it to come closer to true cardiac physiology as well as validation by large cohort studies will establish the method in routine practice. A direct comparison between GLS and MW in different cardiac diseases and the association of this new marker with the prognosis of patients would enhance its role in LV assessment. Treatments that are guided by GLS evaluation, such as in cardio-oncology patients, may benefit most by a more accurate evaluation of the toxic results of chemotherapeutic drugs. Research will prove the effectiveness of this method.

Finally, it has to be mentioned that MW is a diagnostic tool exclusively supported by General Electric (GE) machines. This limits the number of patients that can be evaluated with this method and restricts the ability to compare the results of the same patient with products from different vendors.

## 12. Conclusions

MW is a feasible, reproducible, and thus reliable method for noninvasive measurement of LV performance. Clinical indications of this method have expanded rapidly in the light of some studies demonstrating the correlation of MW with invasive stroke work and its role as a prognostic marker for survival and hospitalizations, especially in HF patients. Because there are several limitations to this method, it is recommended to be used above and beyond EF and GLS for routine cases or as a research tool. Larger multicenter studies that include a broad spectrum of cardiac diseases will determine the added value of MW in LV performance assessment.

## Figures and Tables

**Figure 1 diagnostics-11-00573-f001:**
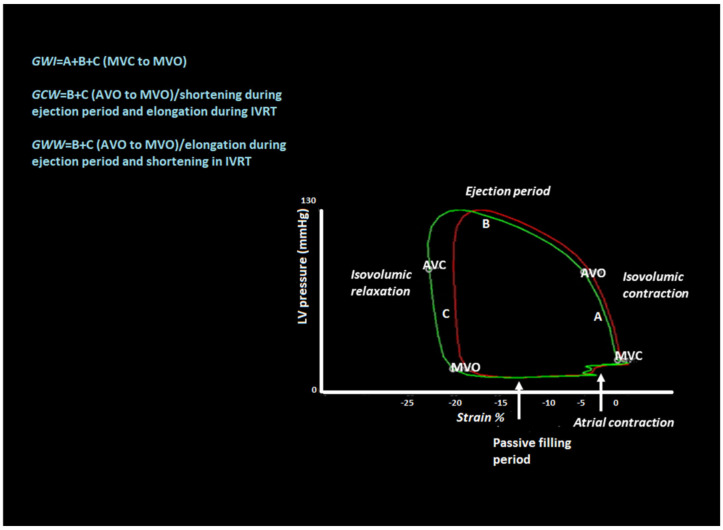
Pressure–strain loop (PSL) with valvular event timing and cardiac cycle phases. Global work index (GWI) represents the area of the loop from mitral valve closure (MVC) to mitral valve opening (MVO). Global constructive work (GCW) reflects the work that contributes to pump from aortic valve opening (AVO) to MVO. There is shortening during the ejection period and lengthening during isovolumic relaxation (IVRT). Global wasted work (GWW) represents the work that does not contribute to ejection. There is elongation of myocytes during the ejection period and shortening after the aortic valve closure (AVC) during IVRT.

**Figure 2 diagnostics-11-00573-f002:**
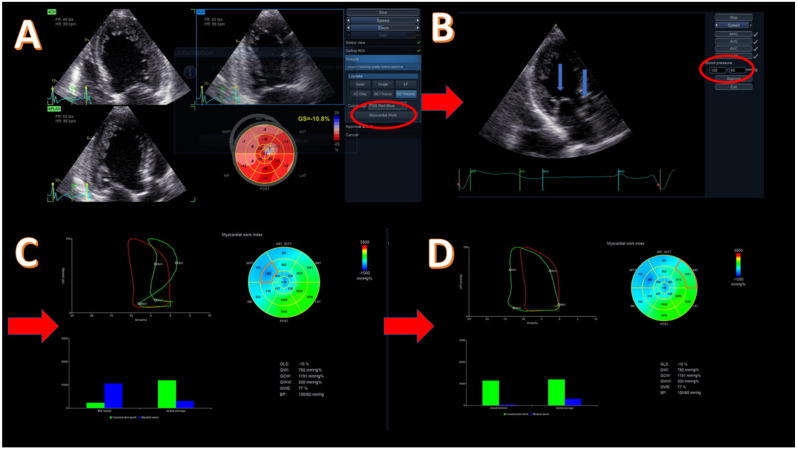
Methodology of myocardial work (MW) for a patient with reduced ejection fraction (EF = 30%) due to cardiotoxicity. (**A**) Acquisition of apical four-, two-, and three-chamber view and evaluation of global longitudinal strain (GLS) with left ventricular (LV) bull’s eye; GLS = −10.8%. The highlighted button (with red circle) moves to the next step for myocardial work evaluation. (**B**) Introduction of systolic and diastolic blood pressure (red circle) and confirmation of the correct event timing in apical three-chamber view. The blue arrows point to mitral and aortic valves that should be clearly demonstrated. (**C**) Evaluation of global values of all MW components, including bull’s eye of the work index, global pressure–strain loop (red loop), and segmental pressure–strain loop (green loop), showing a dyssynchronous mid-septum segment and bars showing the relationship between the constructive (green bar) and wasted (blue bar) segmental (left) and global work. (**D**) Evaluation of the same global values as in image C but with different segment for pressure–strain loop analysis and with constructive/wasted work bars. Basal anterior segment with constructive work similar to the global one. Red arrows show the next step of the methodology.

**Figure 3 diagnostics-11-00573-f003:**
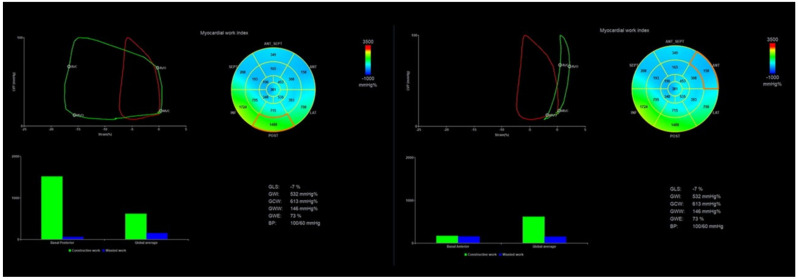
Patient with ischemic cardiomyopathy. Red loop is equal to the global MWI of the patient, while green loops show the PSL of a normal and abnormal segment of LV. GLS = −7%, GWI = 532 mmHg%, GCW = 613 mmHg%, GWW = 146 mmHg%, GWE = 73%. Left green and blue bars of every screen represent the ratio of segmental constructive and wasted work, while bars at the right side represent their global values. You can appreciate the huge difference in the segmental values between the basal posterior segment (**left** screen) and the basal anterior segment (**right** screen) in a patient with extensive anterior myocardial infarction. The basal posterior creates a large pressure–strain loop and constructive work and contributes to ejection, while the basal anterior shows equal levels of constructive and wasted work.

**Figure 4 diagnostics-11-00573-f004:**
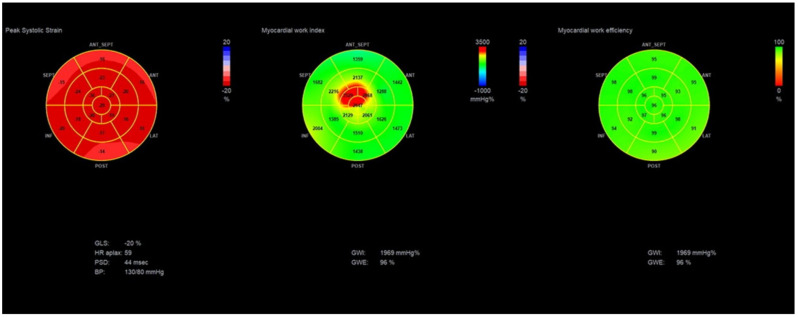
Bull’s eye of GLS, GWI, and GWE showing global and segmental values in a normal heart. GLS = −20%, GWI = 1969 mmHg%, and GWE = 96%.

**Figure 5 diagnostics-11-00573-f005:**
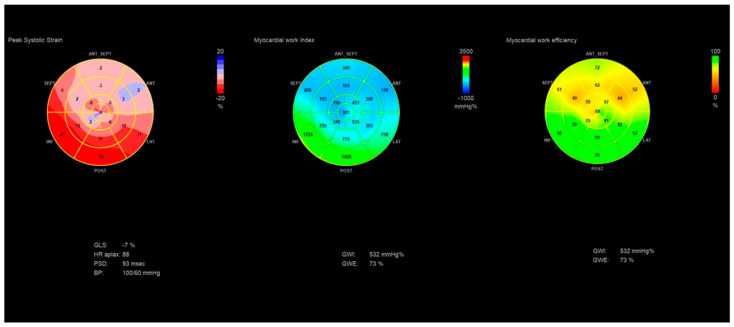
Bull’s eye of GLS, GWI, and GWE showing global and segmental values in a patient with ischemic cardiomyopathy. GLS= −7%, GWI = 532 mmHg%, GWE = 73%.

**Table 1 diagnostics-11-00573-t001:** Analysis of myocardial work components.

Variable	Meaning
Global work index (GWI)	Total work; the PSL area calculated from mitral valve closure to mitral valve opening
Global constructive work (GCW)	Total work contributing to pump function; shortening of the myocytes during systole and lengthening during isovolumic relaxation
Global wasted work (GWW)	Work that does not contribute to ejection; elongation of myocytes during systole and shortening against a closed AV
Global work efficiency (GWE)	Fraction of GCW to GCW + GWW

**Table 2 diagnostics-11-00573-t002:** Reference values in healthy subjects according to three different studies.

	Total Mean ± SD or Median (IQR)	Total Mean ± SD or Median (IQR)	Total Mean ± SD or Median (IQR)	Total Mean ± SD or Median (IQR)
Variables	GWI (mmHg%)	GCW (mmHg%)	GWW (mmHg%)	GWE (mmHg%)
NORRE study	1896 ± 308	2232 ± 331	78.5 (53–122.2)	96 (94–97)
STAAB cohort	2209 ± 307	2430 ± 351	74 (54–101)	96 (95–97)
Galli et al.	1926 ± 247	2224 ± 229	90 (61–123)	96 (94–97)

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
