# Peer review of "Myocardial Work: Methodology and Clinical Applications"

_diagnostics, 2021, doi:10.3390/diagnostics11030573_

Round 1
Reviewer 1 Report
The authors provide an extensive review of the papers that used a novel echocardiography index: myocardial work (a measure of global and regional left ventricular performance that incorporates—besides systolic longitudinal strain—cuff-measured systolic blood pressure and systolic time intervals from echocardiography as a representation of afterload).
The paper reviews all the available literature on various cardiac conditions. However, there is no critical approach to the topic, and conclusions may be misleading for the readers as the authors seem to have limited knowledge of the topicy.
Some of the issues related to the acritical use of the echocardiographic estimation of myocardial work in comparing different patients can be obtained from 10.1016/j.echo.2020.10.007
In addition, there are a few additional issues that should be considered by the Authors :
- the approximation of the brachial cuff blood pressure to LV pressure loop does not take into consideration that: 1. The method does not estimate the diastolic pressure curve, so this has to be assumed normal; and 2. Pulse wave reflection is not taken into account because only the cuff BT is used for calibration of SBP
- The pressure-strain loop is an index, but not the exact equivalent of the pressure-volume loop as stated by Russel et al in their seminal paper (ref 6 and 7 of the submitted paper). Using global longitudinal strain and LV pressure gives a global pressure-strain loop; a proxy of true global myocardial work. However, longitudinal shortening (i.e. global longitudinal strain) contributes only a fraction to SV, this global myocardial work proxy should be a fraction of the true global myocardial work, although compensated somewhat by speckle-tracking derived strain overestimating global longitudinal strain somewhat. (e.g in patients with infiltrative myocardial diseases a large part of the stroke volume is supported by transversal displacement and the echocardiographic global myocardial work will be significantly lower than the actual one.
- Applying estimated LV pressure to different segments it is unlikely to add information, simply because the pressure curve is the same for all segments or walls, so the different strain-pressure loops arise from plotting different strain curves against the same pressure curve, the differences lie in the strain.
- How can myocardial work estimated by echocardiography be independent on afterload when it is a product of strain and blood pressure? See the paper 10.1111/jch.14053 which showed that hypertensive patients (higher BP) and hypertensive + diabetes patients have higher global myocardial work index and constructive work, but they had a worse prognosis compared to controls. Is higher MW dangerous?
Author Response
Reviewer 1
Dear reviewer,
Thank you for your comments. We made a lot of changes according to them and corrected all lingual mistakes. In fact you will appreciate that the revised manuscript contains our perspective-critique about this method in different cardiac conditions, limitations and future directions of MW and some new references that have been published recently. We hope the revised version is eligible for publication.
The paper reviews all the available literature on various cardiac conditions. However, there is no critical approach to the topic, and conclusions may be misleading for the readers as the authors seem to have limited knowledge of the topicy.
Critical approach has been adopted througout the whole manuscript and especially in the «limitations and future directions» section. Conclusions have been changed according to the current status of the method. We hope that the manuscript is written now in the right way.
Some of the issues related to the acritical use of the echocardiographic estimation of myocardial work in comparing different patients can be obtained from 10.1016/j.echo.2020.10.007
This reference has been taken into account and data from it were included in our manuscript.
In addition, there are a few additional issues that should be considered by the Authors :
- the approximation of the brachial cuff blood pressure to LV pressure loop does not take into consideration that: 1. The method does not estimate the diastolic pressure curve, so this has to be assumed normal; and 2. Pulse wave reflection is not taken into account because only the cuff BT is used for calibration of SBP
Indeed 1) LV diastolic pressure is not in the equation of this method because as in stroke work, in myocardial work as well LVEDP is considered very low, approx 2-3mmHg and does not affect the work. 2) pulse wave reflection is important in aortic stiffness as it increases the afterload. This is not represented by the SBP and it is a limitation of the method. We have included in our manuscript that next versions should take into consideration pulse wave velocity, Zca or Zva index or other ways to include afterload from arterial stiffness.
- The pressure-strain loop is an index, but not the exact equivalent of the pressure-volume loop as stated by Russel et al in their seminal paper (ref 6 and 7 of the submitted paper). Using global longitudinal strain and LV pressure gives a global pressure-strain loop; a proxy of true global myocardial work. However, longitudinal shortening (i.e. global longitudinal strain) contributes only a fraction to SV, this global myocardial work proxy should be a fraction of the true global myocardial work, although compensated somewhat by speckle-tracking derived strain overestimating global longitudinal strain somewhat. (e.g in patients with infiltrative myocardial diseases a large part of the stroke volume is supported by transversal displacement and the echocardiographic global myocardial work will be significantly lower than the actual one.
PSL and MW is an index and not the actual LV stroke work. The method does not involve work from circumferential direction and does not into account geometry. It has been stated clearly into the manuscript that MW is not Stroke Work. It is a more accurate method to estimate the segmental work in different situations. Global values however seem to correlate well with invasive measurements but still we can only use it for research and we are at the beginning. All these thoughts and considerations about the method have been analyzed into the manuscript.
- Applying estimated LV pressure to different segments it is unlikely to add information, simply because the pressure curve is the same for all segments or walls, so the different strain-pressure loops arise from plotting different strain curves against the same pressure curve, the differences lie in the strain.
We agree with this comment that BP is the same for all segments and differences lie on the strain. This is a strain-derived method for estimating segmental work. The calculation of MW though is not simply multiplying strain x SBP as the value comes from the PSL area. For example, longitudinal strain will not be able to differentiate a dyssynchornous segment from a hypokinetic or akinetic. In dyssynchony, segmental work might be negative due to high value of wasted work. This will not be reflected by strain x SBP. Further validation of the method has to be done in our opinion.
- How can myocardial work estimated by echocardiography be independent on afterload when it is a product of strain and blood pressure? See the paper 10.1111/jch.14053 which showed that hypertensive patients (higher BP) and hypertensive + diabetes patients have higher global myocardial work index and constructive work, but they had a worse prognosis compared to controls. Is higher MW dangerous?
MW is not independent on afterload but is is adjusted for afterload. A meta-analysis of studies with GLS showed that values are decreased when afterload increases and this does not reflect the LV contraction. By incorporating SBP into the protocol, you eliminate the affect that afteload has on strain segmental values. That way MW is considered «less load-dependent».
About paper 10.1111/jch.14053 we would first like to comment that we weren’t able to see any prognostic information for MW. The authors did not associate MW with survival or hospitalizations, they just calculated and compared controls, hypertensive patients and hypertensive+diabetic patients and demosntrated that in hypertension MW indices increase. It is expected to have increased values of MW in hypertension as the energy that all segments must provide in order to overcome the afterload is raised. In chonic situations, pressure overload produces LV remodeling, tissue fibrosis and poor prognosis, and then strain values and MW decrease. So the answer would be that MW will increase in hypertensive patients temporarily reflecting the extra work that LV needs to produce. Throughout the years, these ventricles will be more easily «exhausted». Higher MW is not dangerous, but it shows that LV strives to overcome the afterload.
Reviewer 2 Report
This manuscript represents a sort of short review about a technological advance that may have an important role in echocardiography in the next years. The authors started from basics and gradually build up the presentation, including several examples that illustrate what unique information it adds and where it can be incremental. When discussing previously published original articles, it is preferable to include some critique on what was not shown and what can hamper the application of the new knowledge. The use of illustrative figures for the clinical applications is strongly encouraged. The concept of non-invasively measured ‘myocardial work’ should be better explained. Please find my comments below:
- One of my main concerns is that the authors consistently state that MW is load-independent. Myocardial work is thought to be less load-dependent compared to ejection fraction or longitudinal strain but its load-independency it is a misconception. This novel method has the advantage over strain that it also takes loading conditions (i.e pressure) into account, but it is not fully loading independent.
- Another important issue that needs to be specified is that the non-invasive method (also used by GE/EchoPAC) was only validated for dyssynchrony. The use of MW for other applications was not validated yet.
- I think the readers are interested whether the technique is available on all echocardiography machines (to my knowledge only GE implemented the non-invasive measurement of myocardial work).
- The authors should as well specify that the non-invasive method used by GE/EchoPAC does not take into account morphology, which is necessary to calculate the ‘true’ myocardial work. The use of pressure as a substitute for force means that wall thickness and radius of curvature are not included, so future development should include LV geometry, wall thickness, and radius in order to measure wall stress instead of pressure. The incorporation in a 3D software could also be of help.
- The authors should write about the limitations of the method and future directions.
- Please correct “were” (written a couple of times instead of ‘where’, for e.g ‘to healthy subjects were GWI increased by 54%...’, or ‘case reports were patients with normal EF and GLS…’)
- Please correct Capital ‘S’ at ‘segmental differences of CW were observed…`
- AS/MI/HFrEF – common abbreviations, but they should be linked to their meaning (Section ‘Valvular heart disease’).
- ‘Similar results were observed in a study with cardiac amyloidosis patients’ – reference is missing.
Author Response
Reviewer 2
Dear reviewer,
Thank you for your kind comments. They were really helpful for our manuscript. We addressed all your comments and queries and we made a lot of lingual correction. We hope the manuscript now is eligible for publication.
- One of my main concerns is that the authors consistently state that MW is load-independent. Myocardial work is thought to be less load-dependent compared to ejection fraction or longitudinal strain but its load-independency it is a misconception. This novel method has the advantage over strain that it also takes loading conditions (i.e pressure) into account, but it is not fully loading independent.
Thank you for your comment. You are absolutely right and we changed this phrase into the manuscript. We also included our perspective about the role of preload, LVEDP and other reasons for increased afterload (e.g. aortic stiffness) that are not reflected in systolic blood pressure. MW is less load-dependent and kind of adjusted for the afterload, so it is not affected like GLS when afterload is increased.
- Another important issue that needs to be specified is that the non-invasive method (also used by GE/EchoPAC) was only validated for dyssynchrony. The use of MW for other applications was not validated yet.
We clearly included in our manuscript that MW was at first introduced by Russel et al and it was validated for patients with dyssynchony and patients with ischemic heart disease. For all other cardiac conditions MW is under research.
- I think the readers are interested whether the technique is available on all echocardiography machines (to my knowledge only GE implemented the non-invasive measurement of myocardial work).
This exclusivity for GE machines has been added in a new paragraph into the manuscript, «Limitations and future directions»
- The authors should as well specify that the non-invasive method used by GE/EchoPAC does not take into account morphology, which is necessary to calculate the ‘true’ myocardial work. The use of pressure as a substitute for force means that wall thickness and radius of curvature are not included, so future development should include LV geometry, wall thickness, and radius in order to measure wall stress instead of pressure. The incorporation in a 3D software could also be of help.
Thank you for this comment. It was very hepful for us to enrich the manuscript. In the text we describe this limitation of the method considering LV architecture, curvature, thickness and radius and the wall stress that is applied on the segments. We also included a comment about 4D echocardiography. We hope you will be covered by our additions.
- The authors should write about the limitations of the method and future directions.
A new extended paragraph has been included into the manuscript
- Please correct “were” (written a couple of times instead of ‘where’, for e.g ‘to healthy subjects were GWI increased by 54%...’, or ‘case reports were patients with normal EF and GLS…’)
These lingual mistakes have been corrected
- Please correct Capital ‘S’ at ‘segmental differences of CW were observed…`
This mistake has been corrected
- AS/MI/HFrEF – common abbreviations, but they should be linked to their meaning (Section ‘Valvular heart disease’).
We analyzed all abbreviations into the manuscript. Including these that you mention.
- ‘Similar results were observed in a study with cardiac amyloidosis patients’ – reference is missing.
There is a reference for cardiac amyloisodis patients and it is included into the manuscript and the references list

Round 2
Reviewer 1 Report
I would like to thank the Authors for having addressed all my comments. The paper has improved. It is more balanced and clinically sound
Reviewer 2 Report
Thank you for having considered the comments made during the first round of revision. Authors did a real effort to answer the questions raised by reviewers.